# Properties of Cold-Bonded and Sintered Aggregate Using Washing Aggregate Sludge and Their Incorporation in Concrete: A Promising Material

Hakan Özkan [1,2,*], Nihat Kabay [2] and Nausad Miyan [3]

1 Oyak Cement Concrete Paper Group/Cimpor Serviços SA, 1099-020 Lisbon, Portugal
2 Department of Civil Engineering, Yildiz Technical University, 34220 Istanbul, Turkey; nkabay@yildiz.edu.tr
3 Department of Civil and Environmental Engineering, Technical University of Darmstadt, 64287 Darmstadt, Germany; nausad.miyan@stud.tu-darmstadt.de
* Correspondence: hozkan@cimpor.com; Tel.: +351-91109-1195

**Abstract:** The aggregate makes up about 65–75% of the total volume of concrete and the use of artificial aggregates manufactured from waste and by-product materials, as an alternative to natural aggregate, has attracted considerable research interest. Washing aggregate sludge (WAS) is obtained as a waste during the process of washing the aggregates, which is disposed or used as landfill. The utilization of WAS as a major component to manufacture artificial aggregates remains unexplored. Therefore, the focus has been directed towards the production of cold-bonded and sintered aggregates using WAS and their incorporation in concrete. The fresh pellets were manufactured using WAS, ground granulated blast furnace slag (GGBFS) and ordinary Portland cement (OPC) and kept in the laboratory conditions at 20 ± 2 °C and 95 ± 5% relative humidity to obtain cold-bonded aggregates, whereas WAS and GGBFS were utilized to manufacture sintered aggregate by heating the fresh pellets up to 1150 °C. The manufactured aggregate properties were characterized through physical, mechanical, chemical, and microstructural analysis. Concrete specimens were also produced by introducing the artificial aggregates in replacement with the coarse aggregate. The results showed that the concrete containing artificial aggregates can be produced with lower oven-dry density and comparable mechanical properties to efficiently utilize WAS.

**Keywords:** washing aggregate sludge; cold-bonded aggregate; sintered aggregate; concrete; compressive strength

## 1. Introduction

Concrete is the most widely utilized material in the construction industry throughout the world [1,2]. Cement, natural sand, coarse aggregate, and water are the major ingredients of concrete. Natural resources of sand and aggregates are progressively diminishing due to the tremendous growth of construction activities across the world [3,4]. Therefore, there is an urgent need to identify alternative options to either replace or minimize the utilization of these aggregates. The construction industries have been facing significant challenges for many years to build sustainable and environmentally friendly structures. The rapid growth of the construction industry has also resulted in the generation of waste associated with construction. In this respect, researchers have been exploring the applications and utilization of industrial by-products and wastes to achieve desired sustainability [5–7].

Washing aggregate sludge (WAS) is a silty clay waste material [8] generated during the classification of crushed aggregates [9]. In other words, the very fine particles (<63 μm) generated during production of crushed sand are removed to improve the quality of sand/aggregate by washing aggregates in an aggregate washing system and the remanent of those aggregates in cake form known as WAS. Approximately 1 million tons of crushed

sand were processed through the aggregate washing system which resulted in the generation of about 100,000 tons of waste WAS, as per the data obtained from the OYAK Cement Concrete and Paper Company located in Cendere, Turkey. This significantly generated WAS is either disposed or landfilled, which consequently creates both environmental and economic problems.

In the last few years, utilization of industrial by-products by blending or partially replacing ordinary Portland cement (OPC) has become popular [10]. However, the utilization of these by-products in the production of artificial aggregate might be an alternative for the consumption of the waste materials since the aggregate volume makes up approximately 65–75% of the total concrete volume [11]. Furthermore, the rapid development and frequent utilization of construction materials have increased the demand for aggregate, cement, and concrete production [12,13]. Artificial aggregates are generally manufactured employing two processes: cement-based granulation (cold-bonding) and firing at high temperature (sintering) [14,15]. In the cold-bonding process, waste material or by-product becomes a water-resistant material with low compressive strength at the initial stage and gains strength depending on the curing method and age [16]. On the other hand, the sintering method, which is mainly based on atomic diffusion, is a common application for mass production of lightweight aggregates without the requirement of long-term curing periods [14]. Cheeseman et al. [17] stated that artificial aggregates should have high strength and low density, and the aggregates should bond strongly with the cement matrix. Generally, sintered aggregates are manufactured using fly ash due to its massive production of approximately 15 million tons per annum from a wide variety of industries [14,18]. The fly ash-based sintered aggregate is generally manufactured at the sintering temperature ranging between 900 and 1200 °C [19]. The properties of artificial aggregates are significantly affected by the binder types and amount [14]. Studies on utilization of mining wastes [20,21], sewage sludge [22,23], WAS [22], different types of ashes [24], and natural materials [25] have been reported in the literature to manufacture artificial aggregates. However, in most of these studies, few of them considered the waste material as the primary source material for artificial aggregate production.

Only few studies regarding the utilization of WAS to manufacture artificial aggregates have been reported in the literature. González-Corrochano et al. [8] manufactured sintered aggregates using WAS and sewage sludge at sintering temperatures between 1175 and 1275 °C for various durations ranging between 1 and 30 min. The produced sintered aggregates had the highest strength of about 3.03 MPa, and maximum dry particle density of 1.48 g/cm$^3$. González-Corrochano et al. [22] investigated various properties of artificial aggregates produced with WAS, sewage sludge, and clay-rich sediment. The authors stated that the artificial aggregate containing 50% WAS and 50% clay-rich sediment achieved the highest compressive strength, lowest water absorption and density. Furthermore, González-Corrochano et al. [26] studied the physical, mechanical and mineralogical properties of sintered aggregate manufactured with WAS, FA and used motor oil.

To the knowledge of the authors, no studies have been reported about the investigation on cold-bonded aggregate using WAS, OPC, and GGBFS and sintered aggregate containing WAS and GGBFS along with their incorporation in concrete. In the present study, physical and mechanical properties of aggregates manufactured using WAS, GGBFS, and OPC were investigated. Furthermore, the produced aggregates were partially replaced in different ratios with coarse aggregate to manufacture concrete. Compressive strength, oven-dry density and water absorption tests were carried out to determine the effectiveness of the manufactured aggregate in concrete.

## 2. Materials and Methodology

WAS, GGBFS, and OPC were used as raw materials in the present study to manufacture cold-bonded and sintered aggregates. WAS was supplied from the sandstone aggregate quarry of the OYAK Cement Concrete and Paper Company located in Cendere, Istanbul, Turkey, whereas GGBFS and OPC were obtained from the local market. OPC of grade

CEMI 42.5 R was also used as the binder in concrete production. WAS was obtained in cake form, which was moist and contained more than 25% water. WAS was initially oven-dried at 100 °C to eliminate water, followed by grinding into powder. The major compounds in WAS were $SiO_2$, $Al_2O_3$, and $Fe_2O_3$, whereas GGBFS comprised of mainly CaO, $SiO_2$, and $Al_2O_3$.

## 3. Manufacturing of Aggregates

The sintered aggregates were manufactured by using WAS and GGBFS powder in equal mass ratios. The optimum sintering temperature and duration were determined based on the preliminary test results. Similarly, different mix ratios were tested to obtain appropriate mix design for cold-bonded aggregates; consequently, WAS, OPC, and GGBFS were mixed in 50%, 30%, and 20% mass ratios, respectively, to manufacture cold-bonded aggregates. Initially, the powder materials were mixed homogenously and placed in the pelletizing machine. The pelletizing machine was then started, and water was sprayed over the blended powder continuously until the formation of spherical fresh pellets occurred. The pelletizing machine and formation of fresh pellets are demonstrated in Figure 1. The fresh pellets were collected and kept in sealed conditions for 24 h. The hardened pellets were sintered at 1150 °C for a duration of 15 min and kept in a controlled environment (20 ± 1 °C, 95 ± 1% relative humidity) for 28 days to obtain sintered and cold-bonded aggregates, respectively. The appearances of the manufactured cold-bonded and sintered aggregates are shown in Figure 2.

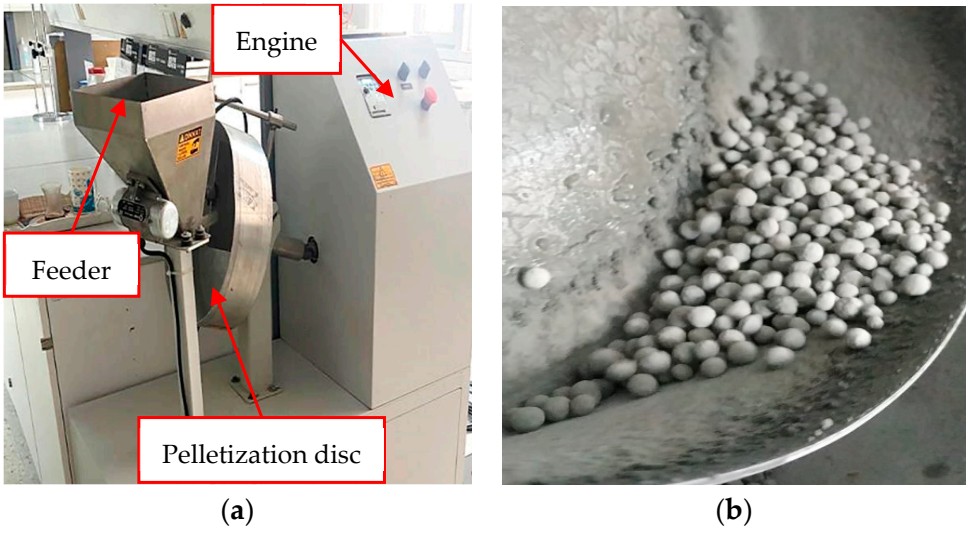

(a)  (b)

**Figure 1.** (**a**) Pelletizing machine and (**b**) formation of fresh pellets.

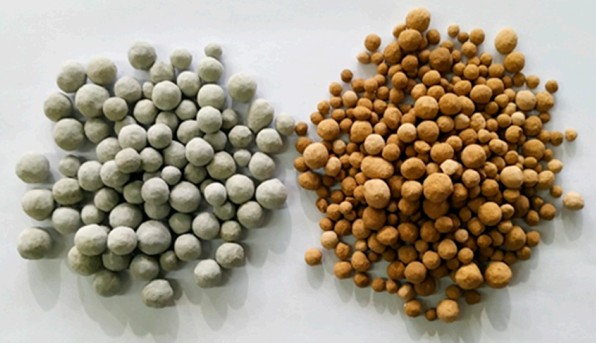

**Figure 2.** The appearance of cold-bonded (**left**) and sintered (**right**) aggregates.

## 4. Test Procedure

### 4.1. Aggregate Tests

The mechanical and physical tests were performed to characterize the manufactured cold-bonded and sintered aggregates. The particle crushing strength (PCS) was determined by placing single aggregate particles between two parallel plates and loading each particle diametrically by using a 28 kN capacity compression machine until failure. A total of 15 aggregate particles were used for the test and the PCS was determined by using the following equation.

$$PCS = 2.8\,F/(\pi d^2)$$

In this equation, F and d represent the failure load (N) and the distance between the loading points (mm), respectively.

Sieve analysis was performed to determine the particle size distribution (PSD) of manufactured aggregates. Chemical tests were carried out for both aggregates to determine sulfate, sulfur, water-soluble chloride salt, water-soluble alkali, and hummus contents in accordance with TS EN 1744-1. The loose bulk density of the manufactured aggregates was determined following the EN 1097-3 standard. Similarly, the water absorption and particle density of the aggregates were determined in accordance with EN 1097-6.

### 4.2. Concrete Tests

The effectiveness of the manufactured aggregates was experimentally investigated in concrete by partially replacing the coarse aggregate in different ratios and by determining the physical and mechanical properties of the concrete. The aggregates used in the reference concrete consisted of a crushed coarse fraction (5–12 mm) and a crushed sand fraction (0–5 mm) obtained from the Cendere quarry, Istanbul. The cold-bonded and sintered aggregates replaced the coarse aggregate in 4 different ratios: 0%, 15%, 30%, and 45% by volume. OPC, CEM I 42.5R, was used as the binder in the production of concrete. A commercially available polycarboxylate formaldehyde-based superplasticizer with a density of 1.07 g/cm$^3$ was used to adjust the workability in all mixes. Potable water was used as the mix water throughout the investigation as well as for the curing of concrete specimens. The water-to-cement ratio was kept constant in all mixes as 0.50. Table 1 shows the mix proportions of the concrete mixtures considering air content of 2% by volume. For mix IDs, "Ref" represents the reference concrete without manufactured aggregates. Similarly, the letters "C" and "S" represent cold-bonded and sintered aggregate-containing concrete, respectively, and the numbers following these letters denote the volume replacement ratios of the corresponding manufactured aggregate with the coarse aggregate. The aggregates were soaked in water for 24 h, and excess water was drained to obtain saturated surface-dried (SSD) conditions prior to concrete mixing. In total, 7 concrete mixes were prepared to systematically investigate the effect of manufactured aggregates on the physical and mechanical performance of the concrete. The compressive strength of the concrete samples was determined according to EN 12390-3 on three 150 × 150 × 150 mm$^3$ cubes at 28 days, and the average was recorded. The water absorption and the oven-dry density of the concrete mixes were determined at 28 days using two replicate disc specimens with dimensions of 100 mm in diameter and 50 mm in height following the ASTM C 642 standard.

**Table 1.** Mix proportions (in kg) for 1 m$^3$ of concrete.

| Mix-ID | Concrete Mixes | | | | | | | |
|---|---|---|---|---|---|---|---|---|
| | OPC | Water | Chemical Admixture | Natural Sand | Crushed Sand | Coarse Aggregate | Cold-Bonded Aggregate | Sintered Aggregate |
| Ref | 360 | 180 | 7 | 355 | 640 | 826 | 0 | 0 |
| C-15% | 360 | 180 | 7 | 355 | 640 | 702 | 88 | - |
| C-30% | 360 | 180 | 7 | 355 | 640 | 579 | 176 | - |
| C-45% | 360 | 180 | 7 | 355 | 640 | 455 | 265 | - |
| S-15% | 360 | 180 | 7 | 355 | 640 | 702 | - | 91 |
| S-30% | 360 | 180 | 7 | 355 | 640 | 579 | - | 183 |
| S-45% | 360 | 180 | 7 | 355 | 640 | 455 | - | 274 |

## 5. Results and Discussion

Table 2 shows the physical and mechanical properties of the manufactured aggregates, and Figure 3 presents the PSD of the manufactured aggregates, natural sand, crushed sand, and coarse aggregate. The oven-dry particle density and loose oven-dry bulk density of both cold-bonded and sintered aggregates were less than 2000 kg/m$^3$ and 1200 kg/m$^3$, respectively, indicating that both manufactured aggregates may be classified as lightweight aggregate according to EN 206 standard. The PCS results indicated that the sintered aggregate had more than 3 times higher strength compared to the cold-bonded aggregate. The water absorption of cold-bonded aggregate was found as 26.4%, about 16% greater than sintered aggregate, indicating that the sintering process enhances the microstructure and results in lower porosity and water absorption compared to the cold-bonding process. Similarly, the particle density of the sintered aggregate was comparatively higher than the cold-bonded aggregate. Figure 3 shows that cold-bonded and sintered aggregates had comparable PSD ranging between 4 and 11.2 mm, and that the manufactured aggregates are slightly finer than the coarse aggregate. Figure 4 shows that the manufactured aggregates are nearly spherical in shape with rough surface texture.

**Table 2.** The physical and mechanical properties of manufactured aggregates.

| Aggregate Types | PCS (MPa) | Water Absorption (%) | Bulk Density (kg/m$^3$) | Particle Density (kg/m$^3$) | |
|---|---|---|---|---|---|
| | | | | Oven-Dry | SSD |
| Cold-bonded | 3.1 ± 0.5 | 26.4 ± 1.0 | 930 ± 55.5 | 1530 ± 45.0 | 1940 ± 65.0 |
| Sintered | 10.0 ± 2.1 | 22.8 ± 1.0 | 940 ± 62.5 | 1640 ± 75.0 | 2010 ± 82.0 |
| Coarse | - | 0.8 ± 0.1 | 1480 ± 60.0 | - | 2710 ± 65.0 |
| Crushed sand | - | 1.3 ± 0.1 | 1500 ± 63.0 | - | 2700 ± 50.0 |
| Natural sand | - | 1.2 ± 0.1 | 1380 ± 70.0 | - | 2620 ± 45.0 |

Table 3 presents the chemical analysis results of the manufactured aggregates. The acid-soluble sulfate and total sulfur contents of cold-bonded aggregate were higher compared to the sintered aggregate, whereas water-soluble chloride and alkali content were similar. Furthermore, the presence of organic matter content was insignificant for both cold-bonded and sintered aggregates. The results indicate that both aggregates satisfy the conditions for their utilization in concrete manufacturing according to the TS EN 12,620 and TS 13,515 standards.

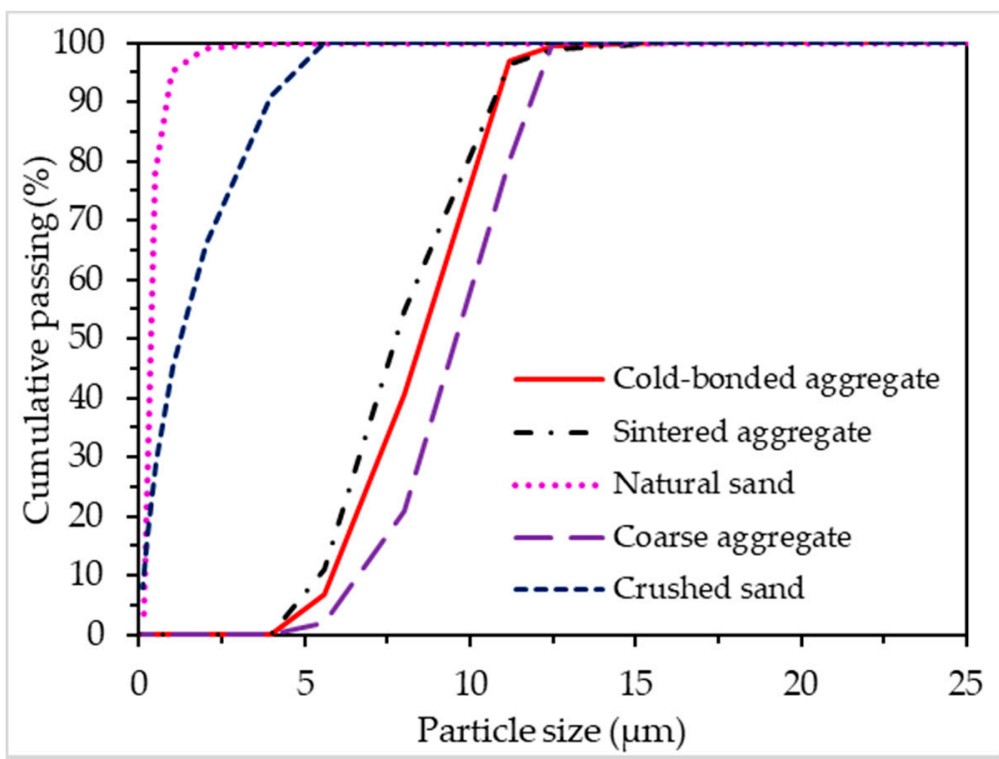

**Figure 3.** Particle size distribution of aggregates.

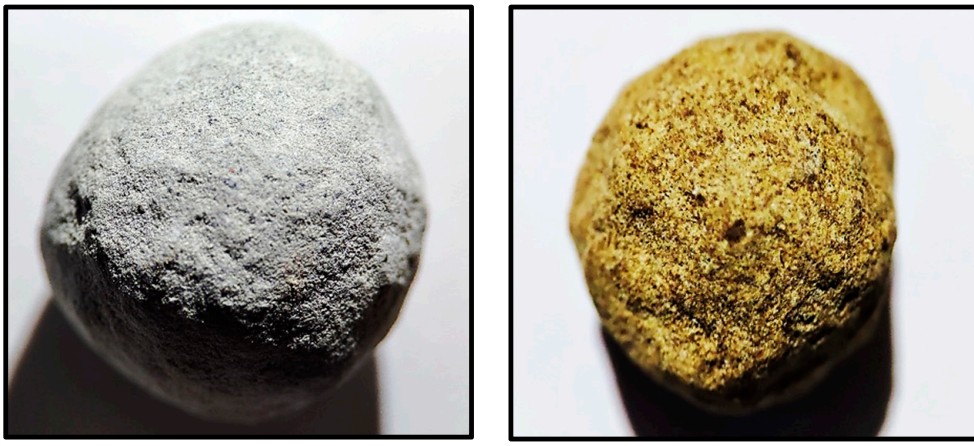

**Figure 4.** Surface texture of the manufactured cold-bonded (**left**) and sintered (**right**) aggregates.

**Table 3.** Chemical analysis and results of cold-bonded and sintered aggregate according to TS EN 1744-1.

| Test | Result | |
|---|---|---|
| | **Cold-Bonded Aggregate** | **Sintered Aggregate** |
| Acid-soluble sulfate content ($SO_3$) | 0.48% | 0.20% |
| Total sulfur content (S) | 0.77% | 0.07% |
| Water-soluble chloride salts ($Cl^-$) | <0.001% | <0.001% |
| Water-soluble alkali content | 0.04% | 0.03% |
| Hummus content | Resulting color was lighter than standard color | |

Table 4 presents the physical and mechanical properties of the investigated concrete mixes. The oven-dry density and the water absorption of the reference concrete were found as 2315 kg/m$^3$ and 5.2%, respectively. The replacement of sintered and cold-bonded

aggregates gradually reduced the oven-dry density (Figure 5), and on the other hand, increased the water absorption (Figure 6). The concrete mixes containing sintered aggregates generally achieved lower oven-dry density and water absorption values for corresponding replacement ratios. For 45% replacement ratio, the water absorption increased by about 54% and 38% in cold-bonded and sintered aggregate-containing concretes, respectively. The significant increase in water absorption can be attributed to the higher water absorption characteristics of the manufactured aggregates compared to the coarse aggregate they replaced. The oven-dry density slightly reduced by 5.9% and 6.7% for concretes containing cold-bonded and sintered aggregates, respectively, compared to the reference concrete.

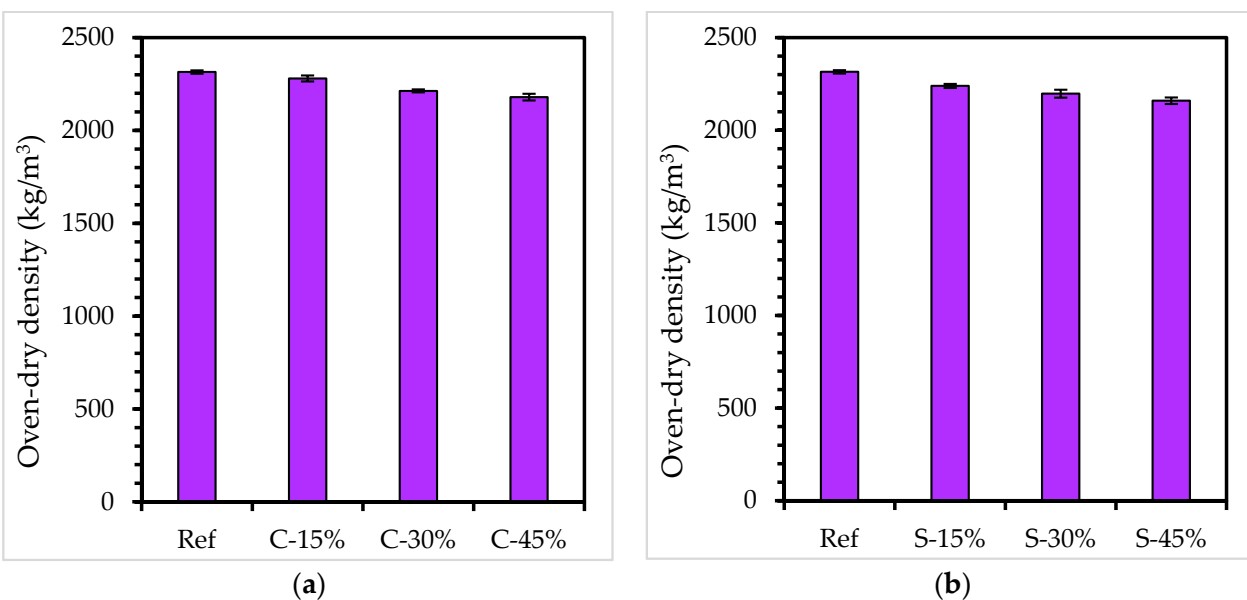

**Figure 5.** Oven-dry density of concrete with: (**a**) cold-bonded aggregate and (**b**) sintered aggregate.

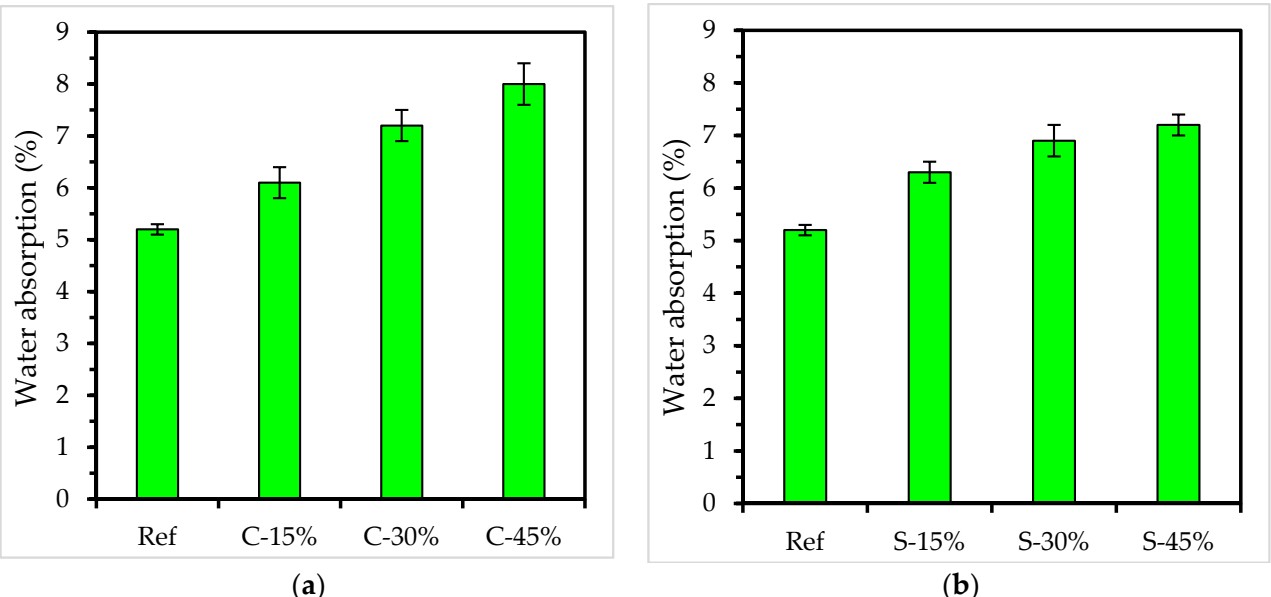

**Figure 6.** Water absorption of concrete with: (**a**) cold-bonded aggregate and (**b**) sintered aggregate.

**Table 4.** Physical and mechanical properties of concrete.

| Mix-ID | Oven-Dry Density (kg/m$^3$) | Water Absorption (%) | Compressive Strength (MPa) |
|---|---|---|---|
| Ref | 2315 ± 9 | 5.2 ± 0.1 | 55.3 ± 1.3 |
| C-15% | 2280 ± 16 | 6.1 ± 0.3 | 57.7 ± 1.8 |
| C-30% | 2213 ± 8 | 7.2 ± 0.3 | 55.7 ± 1.7 |
| C-45% | 2179 ± 18 | 8.0 ± 0.4 | 48.9 ± 1.6 |
| S-15% | 2239 ± 11 | 6.3 ± 0.2 | 52.7 ± 1.6 |
| S-30% | 2197 ± 21 | 6.9 ± 0.3 | 50.8 ± 1.8 |
| S-45% | 2159 ± 18 | 7.2 ± 0.2 | 46.6 ± 1.7 |

Figure 7 shows the 28 days compressive strength of the concrete mixes. The compressive strength of the cold-bonded aggregate-containing concrete ranged between 48.9 MPa and 57.7 MPa, and the strength consistently reduced with the increasing replacement ratios. The compressive strength of the concrete specimens containing 15% of cold-bonded aggregate enhanced by about 4%, whereas the strength was comparable and slightly lower for 30 and 45% of cold-bonded aggregate incorporation, respectively, compared to the reference concrete.

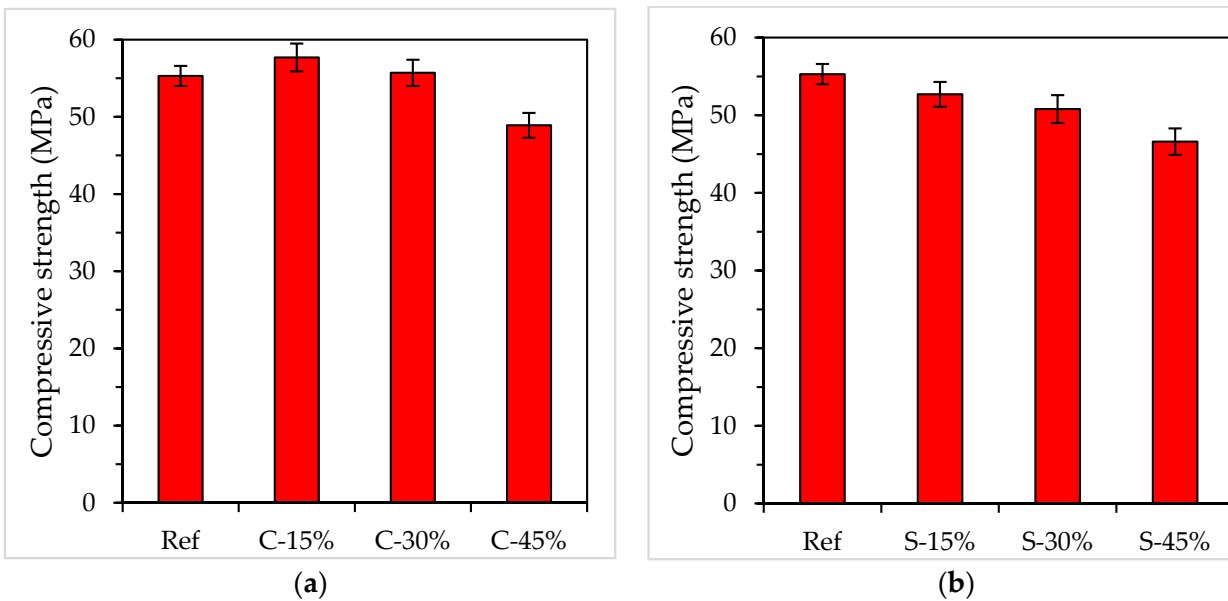

**Figure 7.** Compressive strength of concrete with: (**a**) cold-bonded aggregate and (**b**) sintered aggregate.

The compressive strength of the concrete specimens containing sintered aggregate varied between 46.6 and 52.7 MPa. The compressive strength of the mixes in which sintered aggregates replaced the coarse aggregate consistently reduced with increasing replacement ratios. The highest reduction in strength was about 16% for 45% of sintered aggregate replacement compared to the reference mix.

The concrete containing cold-bonded aggregate achieved higher compressive strength for all corresponding replacement ratios compared to the concrete with sintered aggregate. As previously mentioned, the cold-bonded aggregate had higher water absorption compared to sintered aggregate, which might have contributed to the strength via internal curing [27–29], although the PCS value of cold-bonded aggregate was lower than sintered aggregate.

Figures 8 and 9 show the relationship between oven-dry density and water absorption and compressive strength and oven-dry density of the concrete mixes. The results demonstrate that, the higher the oven-dry density, this results in higher compressive strength and lower water absorption values for the concrete mixes.

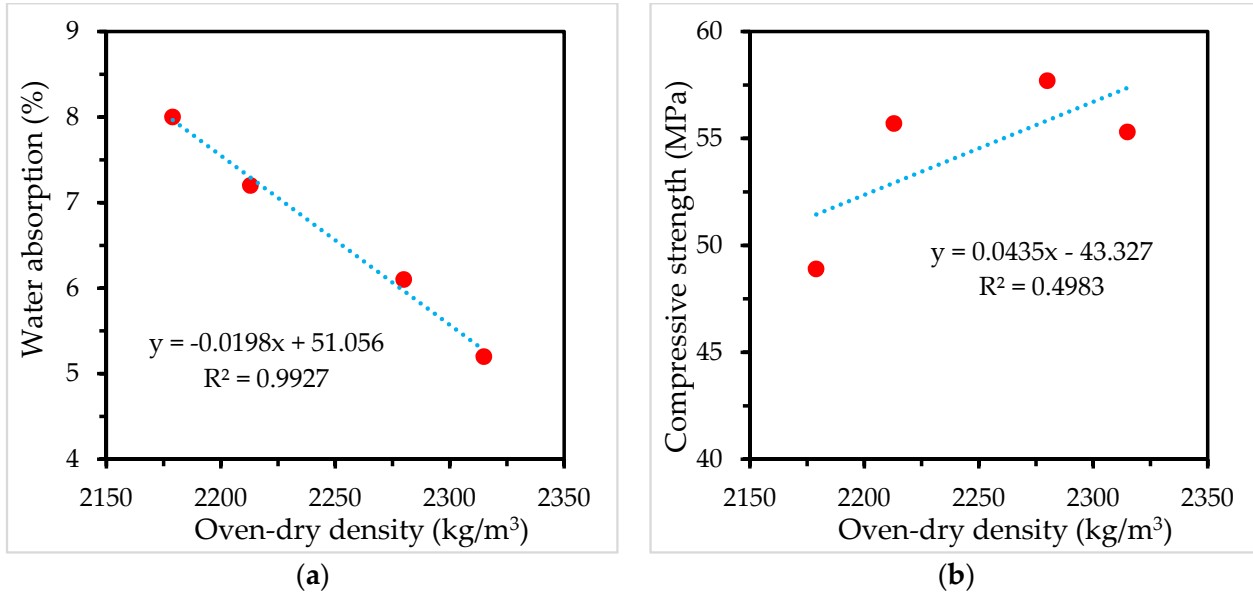

**Figure 8.** The relationship between oven-dry density and (**a**) water absorption and (**b**) compressive strength of cold-bonded aggregate-containing concrete.

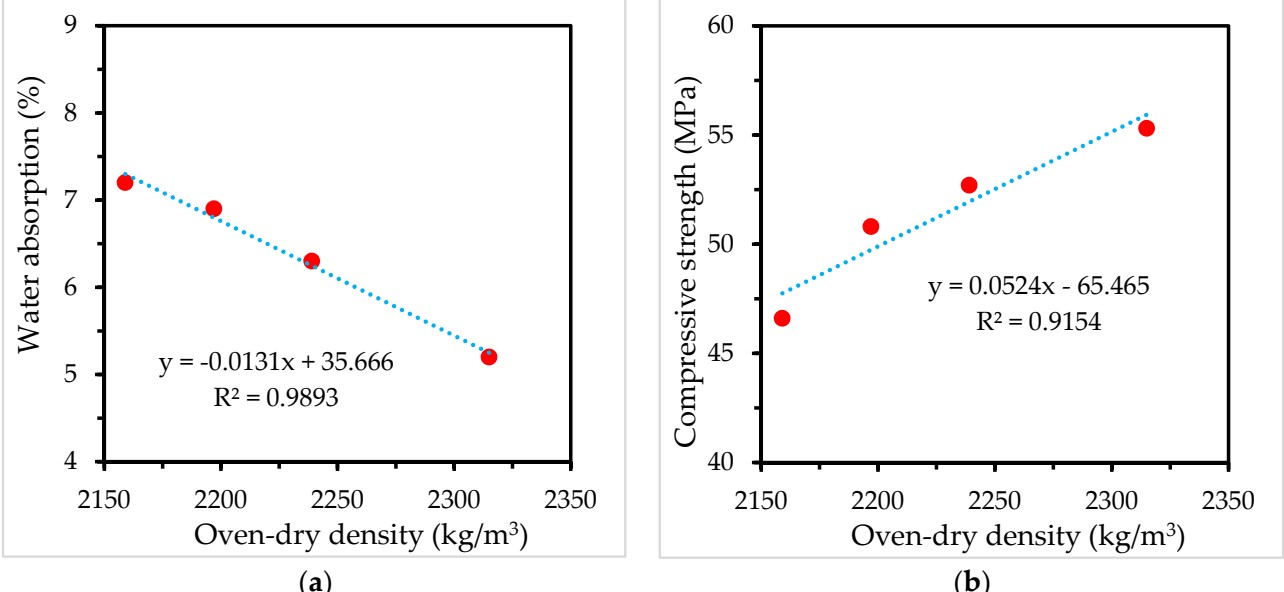

**Figure 9.** The relationship between oven-dry density and (**a**) water absorption and (**b**) compressive strength of sintered aggregate-containing concrete.

Figure 10 illustrates the optical microscopic observations of cold-bonded and sintered aggregate concretes. The distribution of the manufactured aggregates in the concrete matrix are evidently observed. The aggregates are well bonded, with the cement matrix showing no signs of separation, which can be attributed to the irregular surface texture of the manufactured aggregates, as previously shown in Figure 4.

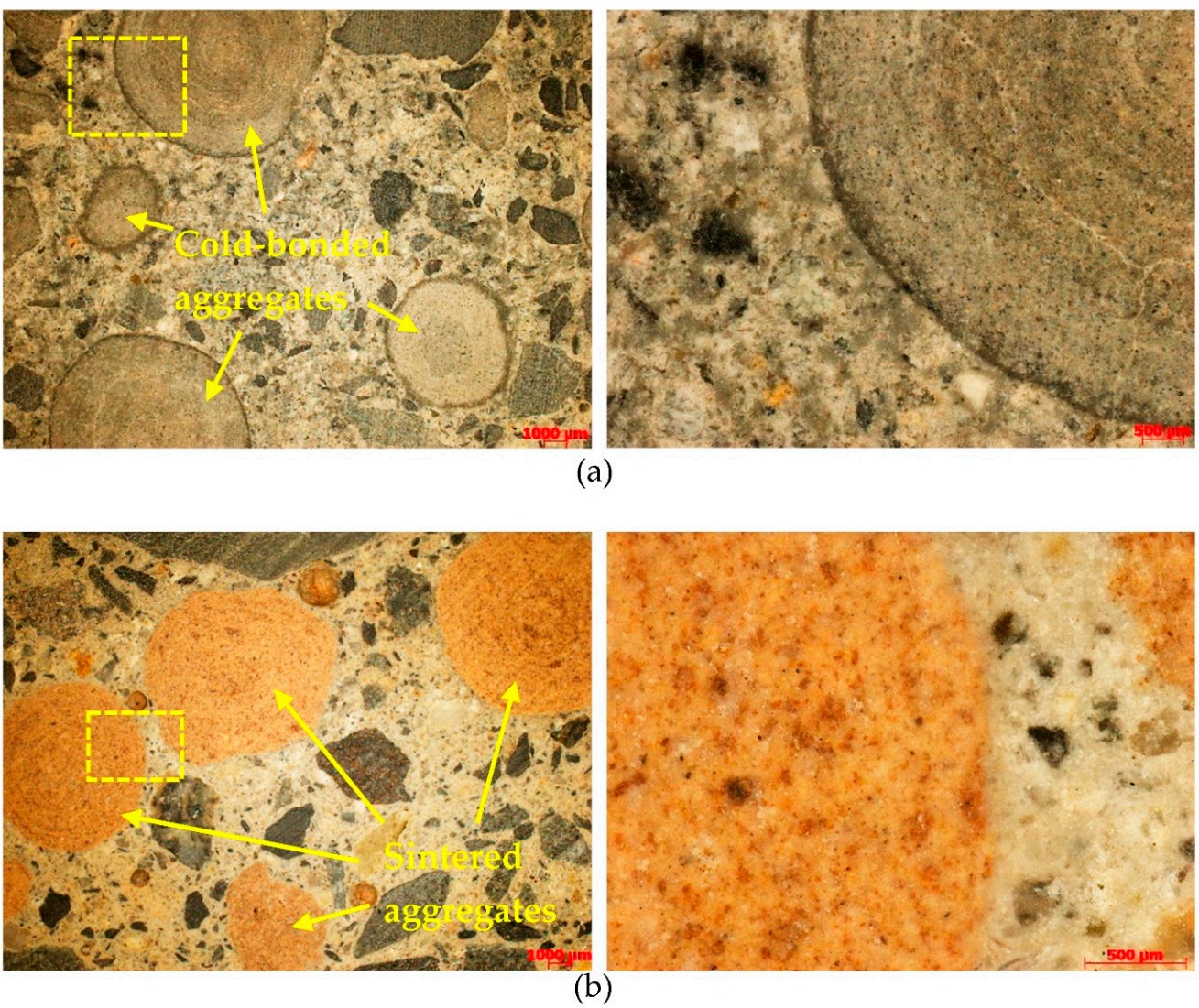

**Figure 10.** Optical microscopic observation of concrete containing: (**a**) cold-bonded aggregate and (**b**) sintered aggregate.

## 6. Conclusions

This paper presents an alternative methodology to utilize the WAS waste generated during classification of aggregates. WAS, OPC and GGBFS were used to manufacture cold-bonded aggregate, whereas sintered aggregate was produced using WAS and GGBFS. Physical, mechanical, chemical, and microstructural properties of the manufactured aggregates were studied. Furthermore, the manufactured aggregates were partially replaced with the coarse aggregate in various ratios to produce concrete, and its physical and mechanical properties were investigated. The following conclusions can be made based on the outcomes of the test results:

(1) The cold-bonded aggregate manufactured with a blend of WAS, OPC and GGBFS in 50%, 30% and 20%, respectively, by mass ratios cured at 20 $\pm$ 1 °C, 95 $\pm$ 1% relative humidity for 28 days had PCS and water absorption values of 3.1 MPa and 26.4%, respectively.

(2) The sintered aggregate containing WAS and GGBFS in equal mass ratios and sintered at 1150 °C for a duration of 15 min attained PCS value of about 10 MPa and water absorption of 22.8%.

(3) The replacement of either aggregate consistently reduced the oven-dry density and increased the water absorption of concrete; however, the performance of sintered aggregate inclusion was better compared to cold-bonded aggregate.

(4) The concrete incorporating 15% of cold-bonded aggregate had approximately 4% higher compressive strength, whereas 30% and 45% cold-bonded aggregate-containing concrete mixes showed similar strength compared to the reference concrete. This enhancement or preservation of strength can be attributed to the internal curing.

(5) The replacement of sintered aggregate by 15%, 30% and 45% reduced the concrete strength by about 5%, 8% and 16%, respectively.

(6) The concrete produced with either cold-bonded or sintered aggregate showed compressive strength greater than 45 MPa at 28 days, which might be appropriate for most civil engineering applications requiring high strength.

(7) The WAS waste generated during washing of aggregate can be efficiently used to produce cold-bonded or sintered aggregate with attractive properties such as particle crushing strength and density. Furthermore, the utilization of WAS-based manufactured aggregate in concrete may be an environmentally friendly approach considering the fact that there is an urgent need to identify the replacement of natural aggregate in concrete. Furthermore, life cycle assessment of concrete containing WAS-based manufactured aggregates should be performed in future studies to investigate economic and environmental benefits.

**Author Contributions:** Conceptualization, N.K. and H.Ö.; methodology, N.K. and H.Ö.; software, N.M. and N.K.; validation, N.K., H.Ö. and N.M.; formal analysis, N.K., H.Ö. and N.M.; investigation, N.K. and H.Ö.; resources, H.Ö.; data curation, N.K. and H.Ö.; writing—original draft preparation, N.K.; writing—review and editing, N.K. and N.M.; visualization, N.K., H.Ö. and N.M.; supervision, N.K.; project administration, N.K. and H.Ö.; funding acquisition, N.K. and H.Ö. All authors have read and agreed to the published version of the manuscript.

**Funding:** This research was funded by the Yildiz Technical University Scientific Research Projects Coordination Unit, grant number FDK-2020-3888, and The APC was funded by CIMPOR Serviços SA.

**Institutional Review Board Statement:** Not applicable.

**Informed Consent Statement:** Not applicable.

**Data Availability Statement:** Not applicable.

**Acknowledgments:** The first author would like to acknowledge that this paper is submitted in partial fulfilment of the requirements for a PhD degree at Yildiz Technical University. The first and second authors would also like to acknowledge the support from the Yildiz Technical University Scientific Research Projects Coordination Unit, grant number FDK-2020-3888.

**Conflicts of Interest:** The authors declare no conflict of interest.

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
