# Peer review of "Properties of Cold-Bonded and Sintered Aggregate Using Washing Aggregate Sludge and Their Incorporation in Concrete: A Promising Material"

_sustainability, doi:10.3390/su14074205_

Round 1

Reviewer 1 Report

Dear Authors,

Thank you for your well-written manuscript, here will be following comments for your consideration:

  1. Abstract has to be shortened (200 words) and novelty of your study has to be clearly underlined in it (like you have done in lines 98-105).
  2. Table 2: please provide densities for your mix components and indicate that volumetric composition of your mix design for 1m3 are ok for all mixes.

Author Response

1) Ans: Revised.

2) Ans: The density values for mix components are presented in Table 1 (OPC), Table 2 (aggregates), and given in the text for the chemical admixture as 1.07 g/cm3. We have rechecked the mix design and the values are ok.

Reviewer 2 Report

Main comment:

The author(s) described that the research novelty of this paper on the lacking of study on artificial aggregates from both OPC and GGBS. However, based on a quick research on the literature on WAS concrete, the search results reviewed that there are past researches on WAS aggregates manufactured with OPC and other wastes such as fly ash. Thus, the author(s) should improve the research background in the Introduction of such related researches, as well as to further refine the research novelty on why the author(s) replace the fly ash with GGBS.

Other than that, based on the authors’ justifications that about 70% of the volume of concrete and the replacements of these coarse aggregates by manufactured WAS aggregates could reduce the environmental impacts from using natural aggregates. However, based on life cycle assessments on the concrete, a full replacement of coarse aggregates by waste materials will impose about 10-25% carbon emission reductions only while the majority of carbon emission of concrete productions is due to the cement content in concrete. In this paper, the author(s) used 30% OPC and 20% GGBS for the manufacturing of the aggregates and this means that higher cement content is used in the mix designs (from both WAS aggregate manufacturing and binder content). Let me take an example, 265 kg/m3 of CAS aggregates is used in the C-45% mix, however, to prepare the CAS aggregates, additional 80 kg/m3 of cement is used. By assuming that 80% of the carbon emission due to cement, the use of 45% WAS aggregates will produce 17.7% more carbon emission due to the additional 80 kg/m3 cement used in aggregate manufacturing, instead of only 9% carbon emission reduction due to coarse aggregates replacements (45%), by assuming that 20% of the carbon emission is due to the coarse aggregates. From my justifications, it seem that manufacturing of WAS using OPC and GGBS might not be a sustainable approach.  The author(s) are advised to answer this concern in order convince the suitability of the paper to the journal.

Other comments:

  • Table 1: The author(s) compared the chemical compositions of WAS, OPC and GGBS in the same table (Table 1), so it will give the impression to the readers that the WAS will be used as cement (or binder) replacements, instead of aggregates. Also, I do not see the reasons that the author(s) compared the chemical compositions, XRD, PSD etc. between WAS and OPC/GGBS, as both are not comparable since they are used as different materials, i.e. aggregates and binder, respectively.
  • Table 2: The reference mix from Table 2 utilized crushed coarse aggregates (5–12 mm) and the size is not within the normal aggregate range which used higher maximum aggregate sizes of 16 to 25 mm. Thus, the results from this paper might not be applicable to the conventional concrete which used higher maximum aggregate sizes.
  • Conclusions: what does the author(s) means on “The WAS waste generated during washing of aggregate can be efficiently used to 344 produce cold-bonded or sintered aggregate with attractive properties”? The attractive properties should be further refined. Also, as I commented above, without a proper environmental assessment, the author(s) cannot conclude that WAS-based manufactured aggregate has positive environmental benefits

Author Response

1) Ans: There are similar studies reported in the literature regarding the manufacturing of sintered aggregate using WAS and fly ash by González-Corrochano et al., however since the content of these papers do not vary much, the authors preferred to refer to the relevant articles.

Fly ash has been previously used with WAS to manufacture sintered aggregate, however WAS and GGBFS has not been studied yet which indicates the novelty of this study. In addition, the effect of WAS-based artificial aggregate on concrete properties has not been investigated yet which might be another contribution to the literature.

2) The authors agree with the reviewer’s comment. The aim of this study was to propose an alternative methodology to utilize WAS waste efficiently, and therefore the target was directed to manufacture aggregate from this waste. From life cycle assessment (LCA) perspective, the cold-bonded aggregate may not be an environmental-friendly approach, which we have not investigated in this study. However, it should be noted that past studies on the cold-bonded aggregates mostly used cement as binder. The authors also have used lower cement ratios and alternative binders such as bentonite and fly ash, however the particle strength of the aggregates were lower in those cases. Therefore, the combination of 50% WAS, 30% Cement, and 20% GGBFS was used to produce cold-bonded aggregates. We also manufactured cold-bonded aggregate with 50% WAS, 10% Cement, and 40% GGBFS, but the strength of the aggregate was almost 1/3 of the presented mix design. The LCA study requires further study using lower cement ratios or alternative methods such as alkali-activation. The last bullet in conclusion has been revised based on this comment.

3) Ans: These are the properties of raw materials used in the study. The aim was rather to characterize the raw materials and no comparison was done among them. The mix proportions of cold-bonded, sintered aggregate and the concrete have been clearly presented in the text.

4) Ans: The manufactured aggregates were obtained in the size range of about 5-12 mm. To maximize the incorporation of the manufactured aggregates the particle size range of the coarse aggregate was also chosen similarly. The authors believe that this will not affect the outcomes of this study. During the pelletization process the size range of the manufactured aggregates might be modified with longer durations and can be used in replacement with coarser aggregates.

5) Ans: Attractive properties have been addressed. The response to the second comment has been previously reported.

Round 2

Reviewer 2 Report

There are two comments/responses that I would like the author(s) to further refine as the responses given are not completely address my previous comments: 

  1. Table 1: Chemical compositions of the WAS are placed together with GGBS and OPC, and this will give the wrong impression to the readers that WAS is used as OPC replacement in this paper. If the author(s) intended to present the "raw material properties", it is suggested that the author(s) present only the direct materials properties, which we directly apply it to the mix designs, i.e. the material properties of the aggregates, not the raw materials that we use to produce the aggregate. In other words, are we interested in the clay that being used to produce the LECA aggregates? The answer is a 'no' as most of the readers are only concerning on the properties of LECA. 
  2.  For the comment on the manufactured aggregates were obtained in the size range of about 5-12 mm. What I meant is that the mix designs in this paper are limited to 12 mm maximum aggregate size only and the results cannot be used to be compared with those references used other maximum aggregates sizes. 

Author Response

The manuscript has been revised according to the reviewer’s comments. Table 1, Figure 1, and Figure 2 are removed from the manuscript to avoid any misunderstanding as recommended by the reviewer and some minor revisions were also made.